# Comparison of Three Common Intervertebral Disc Discectomies in the Treatment of Lumbar Disc Herniation: A Systematic Review and Meta-Analysis Based on Multiple Data

**DOI:** 10.3390/jcm11226604

**Published:** 2022-11-08

**Authors:** Xiao-ming Zhao, An-fa Chen, Xiao-xiao Lou, Yin-gang Zhang

**Affiliations:** Department of Orthopaedics of the First Affiliated Hospital, Medical School, Xi’an Jiaotong University, Xi’an 710061, China

**Keywords:** lumbar disc herniation, percutaneous transforaminal discectomy, microendoscopic discectomy, traditional open surgery, systematic review and meta-analysis

## Abstract

Objective: Due to recent developments and the wide application of percutaneous transforaminal discectomy (PTED), we herein compare it with microendoscopic discectomy (MED) and traditional open surgery (OD) through surgical indicators and postoperative outcomes to evaluate the advantages and disadvantages of minimally invasive surgery PTED. Methods: This systematic review and meta-analysis was conducted in line with PRISMA guidelines (PROSPERO2018: CRD42018094890). We searched four English and two Chinese databases from the date of their establishment to May 2022. Randomized controlled trials and case–control studies of PTED versus MED or PTED versus OD in the treatment of lumbar disc herniation were retrieved. Results: A total of 33 studies with 6467 cases were included. When comparing MED with PTED, the latter had less intraoperative blood loss, smaller incision, shorter postoperative bed times, shorter hospitalization times, better postoperative visual analogue scale (VAS) for low back pain, and postoperative dysfunction index (Oswestry Disability Index, ODI) and higher recurrence rates and revision rates. However, operation times, postoperative VAS leg scores and complications, and successful operation rates were similar in both groups. Comparison of PTED with OD revealed in the former less intraoperative blood loss and smaller incision, shorter postoperative bed times, shorter hospitalization times, shorter operation times, and higher recurrence rates and revision rates. Nonetheless, comprehensive postoperative VAS scores, VAS leg pain scores, VAS low back pain, ODI and incidence of complications, and successful operation rates were similar between the two groups. Conclusions: The therapeutic effect and safety of PTED, MED and OD in the treatment of lumbar disc herniation were comparable. PTED had obvious advantages in that it is minimally invasive, with rapid recovery after surgery, but its recurrence rates and revision rates were higher than MED and OD. Therefore, it is not possible to blindly consider replacing MED and OD with PTED.

## 1. Introduction

Lumbar disc herniation is a public health problem that negatively impacts people’s health. Waist and leg pain caused by this condition has long affected the work and daily life of people from all walks of life [1]. The basic therapeutic principle of lumbar intervertebral disc herniation is that surgical treatment is required under the condition that conservative treatment has provided no significant improvement [2]. With the development of science and progress of science and technology, the surgical method for lumbar disc herniation continues to change. The method of treatment mainly includes open surgery (OD), microendoscopic discectomy (MED) and the most popular and most widely used technique, percutaneous transforaminal discectomy (PTED). The common operation for OD is extirpation of the total laminectomy and semi-laminectomy. In recent years, several techniques have been developed, including posterior lumbar interbody fusion surgery (PLIF) [3], transforaminal lumbar interbody fusion surgery (TLIF) [4], and, on the basis of minor trauma, small minimally invasive transforaminal lumbar interbody fusion surgery (MIS-TLIF) [5], among others. The surgical approach is usually from the back, requiring the removal of part of the lamina and the upper and lower articular processes to fully expose the surgical field of view. The aim of using this approach is to enable the surgeon to clearly remove the protruding disc tissue by sight. The postoperative effect is good, and this approach has been considered by the industry for many years to be the standard surgery for the treatment of lumbar intervertebral disc herniation [6]. MED uses the work channel to decompress the lumbar spinal canal and lateral recess, and the abnormal nucleus pulposus is removed using an endoscope to meet the surgical aim [7,8]. The combination of MED and OD and the operation mode is actually a minimally invasive OD. During the operation, only a small amount of muscle next to the vertebral body is removed, the removed amount of vertebral plate and yellow ligament is very small, and the normal anatomical structure is relatively intact. Fontanella et al. [9] reported that treatment of LDH was good and that the operation success rate was 97%. At one time, MED was considered the “golden standard” for the treatment of lumbar intervertebral disc herniation [10], and many medical institutions at home and abroad are still carrying out this technique [11].

The most popular currently used minimally invasive technique—endoscopic spine surgery—originated in Europe and America [11,12], and has been further developed in east Asia, especially in China. Endoscopic spine surgery is mainly divided into two approaches, the earliest transforaminal approach (PTED), and the later-developed translaminar approach, percutaneous endoscopic interlaminar discectomy (PEID). Compared with PTED, PEID is similar to open surgery and microendoscopic discectomy, which is a more minimally invasive small-incision MED. Its indications are relatively narrow, mainly for lumbar 4–5, lumbar 5-sacral 1 disc herniation, and high iliac crest, i.e., for patients who cannot undergo PTED. Therefore, this study focuses on the more commonly used transforaminal endoscopic system, namely PTED. Compared with traditional open-window surgery, the surgical trauma from this technique is small and postoperative recovery is fast. This technique maintains the stability of the lumbar spine to a great extent [13]. Compared with MED, the operation channel enters the prominent intervertebral space through the natural intervertebral foramen, thus greatly reducing some of the complications caused by the drenching of the dural sac and nerve root. Lumbar disc herniation is a common and frequently occurring disease in the clinic, and PTED is a common surgical procedure for LDH treatment. Although the advantages of the microstructure of PTED are obvious, it has similar therapeutic effects to traditional open surgery and MED [10]; however, its disadvantages are not negligible. The learning curve of PTED is steep [14], making this treatment not suitable for young doctors. PTED is costly and has a high exposure rate. Recently, many related studies [15] have found another fatal flaw of PTED: it greatly exceeds MED and open surgery in regard to the incidence of long-term complications, recurrence rate and postoperative renovation rate. Thus, is PTED really an appropriate replacement of MED and traditional open surgery as a standard procedure for the treatment of lumbar intervertebral disc herniation?

This meta-analysis compared the application of PTED with OD and MED from multiple perspectives to assess the advantages and disadvantages of all types of surgical methods to explore, in detail, whether other lumbar and resection treatments for lumbar intervertebral disc protrusion can be completely replaced with PTED, and to guide clinicians and patient treatment options to provide a better outcome.

## 2. Methods

### 2.1. Protocol and Registration

This systematic review and meta-analysis was registered with the International Prospective Register of Systematic Reviews (PROSPERO: CRD42018094890, http://crd.york.ac.uk/PROSPERO accessed on 8 August 2022).

### 2.2. Inclusion Criteria

Study design: comparison of randomized controlled clinical trials of MED and MED and traditional open surgery for lumbar intervertebral disc herniation, as well as prospective or retrospective controlled studies.Patients: patients were diagnosed with lumbar intervertebral disc herniation, not limited by age or sex, and initial onset or recurrent disease.Intervention: in the case of patients, MED, traditional open surgery, and other interventions; the premise is that these interventions are consistent across groups.Outcomes: the included studies must include at least 3 of the evaluation indicators studied.

### 2.3. Exclusion Criteria

Meeting summaries and meta-analysis summaries, among others, are not included in this study.Incomplete outcome indicators in the literature, such as the average number of results but absence of the standard deviation.No control group in the study, such as case series and other observational studies, among others.Combination of other diseases affecting the curative effect, such as spondylolisthesis and intervertebral disc calcification, among others.

### 2.4. Search Strategy

This meta-analysis was conducted in strict accordance with the PRISMA statement (Preferred Reporting Items for Systematic Reviews and Meta-Analyses) [16]. Through computer retrieval from databases such as PubMed, EMBASE, Cochrane Library, Ovid, China Knowledge Network and Wanfang, all relevant controlled trials were collected, and the retrieval time ranged up to May 2022. In accordance with the principles of evidence-based medical literature retrieval (PICOS) [17], the following main key words were used: lumbar disc herniation, percutaneous transforaminal endoscopic discectomy, percutaneous endoscopic discectomy, microendoscopic discectomy, fenestration discectomy, open discectomy surgery, and controlled clinical trials, its synonyms and close synonyms, among others, to repeatedly retrieve relevant publications. First, by reading the title and abstract, we conducted preliminary screening. Then, after reading the full text again and screening, we determined whether the article was included. The author conducted a manual search for references with more citations and used Baidu Academic, Google and other search engines to find relevant literature on the Internet. We also searched the grey literature [18]. For some valuable and possibly unattainable literature studies, it was necessary to contact the author to obtain information. The detailed search terms are listed in Appendix A.

### 2.5. Literature Screening and Quality Evaluation

Two evaluators [ZXM, LXX] independently screened the qualified literature according to predetermined inclusion criteria. The methodological quality evaluation of the selected literature was evaluated by two independent researchers [ZXM, LXX]. In the case of disagreement, the decision was made by negotiation or by a third researcher [CAF]. The specific evaluation method of literature quality is shown below.

For randomized controlled trials (RCT), the modified Jadad scale [19] was used. Its specific criteria are as follows: randomization method (0~2 points: no narrative, or narrative inappropriate, 0 points; related random but specific methods, 1 point; describe appropriate stochastic methods such as random number tables, or a similar method, 2 points); hidden distribution (0~2 points: narrative, or narrative inappropriate, such as an alternate allocation, 0 points; only shows random number tables or other unclear but specific methods, 1 point; describes the proper allocation concealment as a sealed envelope or similar method, 2 points); blinding method (0~2 points: the method is not double-blinded or blinded, 0 points; the blind method is only described with an unknown number, 1 point; and a description of double-blind, placebo or a similar method, 2 points); loss of visits and exit (0~1 points: no account, 0 points; narrative, 1 point). The total score of this evaluation is 7, and the total score of 4~7 is regarded as high-quality research.

For a retrospective study, the Newcastle–Ottawa Scale (NOS Scale) [20] was used for the evaluation. Its specific criteria are as follows: the researcher discusses the quality of each document according to the standard of NOS, and the decision of final inclusion or elimination of the document is formed after consensus is reached. The NOS scale has eight entries as follows. Selection of the research object has four items, namely the definition of the case and the appropriateness of the diagnosis, the representativeness of the case, selection of the control, and definition of the control, with a total of 4 points; the comparability of the group, which is the comparability of the experimental and the control group in the design and analysis stage, with a total of 2 points; results, comprising three items, namely the exposure investigation and evaluation method, the same method for the case and object investigation, and no response rate of the two groups, with a total of 3 points and a total score of 9. The total score is greater than 6 for high-quality research and less than 6 for low-quality research.

### 2.6. Data Extraction

We extracted the relevant data from different experiments and recorded them on a spreadsheet, including (1) basic information (author, year of publication, research types, operation method, number of follow-up visits, evaluation index); (2) analysis indicators: postoperative therapeutic indexes, such as the postoperative visual analogue scale (VAS) [21] score and Oswestry Disability Index (ODI) [22], complications of surgery (in this indicator, recurrence will not be included in the complications), recurrence and residual rate, rate of good renovation and surgical effects (according to the modified MacNab criteria) [23]; surgical indicators, such as incision length, operation time, intraoperative bleeding, postoperative bed time, and hospital stay. According to the unified design table, two researchers [ZXM, LXX] independently extracted and cross-checked data from the original text and. If there was a disagreement, then discussions were carried out and the decision was made by the third researcher [CAF]. If data were lost or a report was incomplete, the person in charge of the clinical trial was contacted if necessary, and basic information required for the meta-analysis that was as complete as possible was acquired.

### 2.7. Statistical Approach

For categorical data, the odds ratio (OR) and 95% confidence interval (CI) were calculated. For continuous data, when all of the clinical trials for the same curative effect used the same unit of measurement, we calculated the mean difference (MD) and 95% CI; otherwise, the standard mean square error (standard mean difference, SMD) was chosen. The heterogeneity test for each study was conducted by the X^2^ test, with *p* < 0.1 as the test level, and the size of heterogeneity was judged according to I^2^. When *p* < 0.1, I^2^ < 50%, the inclusion study was considered to have greater heterogeneity, and the random-effects model was used for the meta-analysis. Subgroup analysis was carried out to further explore the heterogeneity source.

If the heterogeneity was not significant (*p* > 0.1, I^2^ < 50%), the fixed-effect model was used for analysis. The result of the evaluation index of the meta-analysis, and sensitivity analysis by the jack-knife method [24], by means of removing low-quality, small-sample-size or large-sample-size literature, was in turn used to further investigate the stability of the results. Finally, a funnel plot was used to analyse the degree of publication bias. Revman 5.2 software provided by the Cochrane collaboration (London, UK) was used for statistical analysis of all data, and all results are expressed as 95% confidence intervals (95% CIs).

## 3. Results

### 3.1. Literature Search

A total of 114 relevant articles were searched in accordance with all the methods and references of the proposed retrieval strategy, all of which were in the English literature. By using the tools to find duplicated documents, 13 papers were removed. After reading the title and summary, 50 studies were excluded that were inconsistent with the evaluation purpose, object and intervention measures of the system. After reading the full text, 21 of the remaining 54 articles were excluded that did not meet the inclusion criteria, and the result indicators are not detailed. The obtained papers were analysed again, and finally, 7 RCTs and 26 retrospective studies were obtained. Among them, 18 compared PTED with MED and 17 compared PTED with open surgery. The total number of subjects in the study was 6467, of which 2064 were in the PTED group, 1791 in the MED group and 2612 in the traditional open surgery group. The literature retrieval process is shown in Figure 1. The specific retrieval items can be seen in Appendix A.

### 3.2. Study Characteristics

All of the included studies were published between 1975 and 2021. Three intervening measures were included. The sample sizes ranged from 20 to 1856 individuals (total 6467). The ages of the patients ranged from 12 to 77 years. The follow-up time ranged from 6 to 120 months. The basic characteristics of the included trials are shown in Table 1.

### 3.3. Literature Quality Evaluation

A total of 33 papers were included in this study, 7 of which were randomized controlled clinical trials. Among them, four articles scored 7 points, 7 points, 7 points and 4 points, respectively, in accordance with the Jadad literature quality evaluation standard, thus representing high-quality RCTs. However, three article scores were lower, representing relatively low-quality RCTs. Twenty-six studies were non-randomized controlled retrospective studies. According to the NOS scale, the literature quality was evaluated, and the quality level of the literature was at a high level, which could be followed up. The results of the literature quality assessment are shown in Table 2 and Table 3.

### 3.4. Outcomes

All the results of meta-analysis for surgical related indicators and postoperative curative effect are shown in Table 4.

#### 3.4.1. Average Operative Time (min)

Fourteen articles [13,25,26,27,28,29,30,31,32,33,34,35,36,37] reported the average operation time of the PTED and MED groups. Among them, there were 883 patients in the PTED group and 1542 in the MED group. The study was heterogeneous (*p* < 0.00001, I^2^ = 99%), and the meta-analysis was conducted using a random-effects model. The results showed no significant differences in the average operation time between the two groups (MD = −8.74, −24.15 to 6.66, *p* = 0.27), indicating that PTED and MED had a similar average operation time. The specific results are shown in Appendix A. Thirteen articles [2,6,32,34,38,39,40,41,42,43,44,45,46] reported the average operation time for the PTED and traditional open operation group, including 608 in the PTED group and 682 in the traditional fenestration group. The heterogeneity in each study was high (*p* < 0.00001, I^2^ = 99%). The random-effects model was used for the meta-analysis. There was a significant difference in operation time (MD = −22.50, −41.03 to −3.98, *p* = 0.02), indicating a slightly shorter operation time in the PTED group than the traditional open operation, and the specific results are shown in Appendix A.

#### 3.4.2. Intraoperative Blood Loss (mL)

Seven articles [27,29,31,32,33,34,35] reported the intraoperative blood loss in the PTED and MED groups. Among them, there were 349 patients in the PTED group and 671 in the MED group. The heterogeneity was high (*p* < 0.00001, I^2^ = 99%), and the meta-analysis was conducted using a random-effects model. The results showed that blood loss was less in the PTED group than the MED group (MD = −24.96, −34.12 to −15.81, *p* < 0.00001), and the specific results are shown in Appendix A. Seven articles [6,32,34,38,42,43,47] reported intraoperative blood loss in the PTED group and fenestration group. There were 292 people in the PTED group and 335 in the fenestration group. The study was heterogeneous (*p* < 0.00001, I^2^ = 99%), and a random-effects model was used for the meta-analysis. The results showed that the amount of blood loss in the PTED group was less than that in the traditional fenestration group (MD = −89.29 −111.80 to −66.79, *p* < 0.00001), and the specific results are shown in Appendix A.

#### 3.4.3. Size of the Incision (cm)

Four articles [29,32,33,35] reported the size of the incision in the PTED and MED groups, with 207 and 475 participants, respectively. Heterogeneity was high (*p* < 0.00001, I^2^ = 100%), and the meta-analysis was performed using a random-effects model. The results showed that the incision was smaller in the PTED than the MED group (MD = −1.56, −2.63 to −0.50, *p* = 0.004), and the specific results are shown in Appendix A. Six articles [6,32,35,43,44,47] reported the incision size in the PTED and traditional fenestration group. Among them, there were 284 people in the PTED group and 343 in the traditional fenestration groups. The heterogeneity was high (*p* < 0.00001, I^2^ = 99%), and the meta-analysis was conducted using a random-effects model. The results showed a smaller incision in the PTED than the MED group (MD = −2.79, −3.51 to −2.06, *p* < 0.00001), and the specific results are shown in Appendix A.

#### 3.4.4. Postoperative Bed Rest Time (Day)

Two articles [30,35] reported postoperative bed rest time in the PTED and MED groups, with 110 and 103 subjects, respectively. The heterogeneity was high (*p* < 0.00001, I^2^ = 97%), and the meta-analysis was conducted using a random-effects model. The results showed that the postoperative bed rest time was shorter in the PTED than the MED group (MD = −2.45, −4.13 to −0.76, *p* = 0.004), and the specific results are shown in Appendix A. Three articles [43,44,45] reported the postoperative bed rest time in the PTED group and traditional fenestration group, with 121 and 113 subjects, respectively. The heterogeneity was high (*p* < 0.0001, I^2^ = 91%), and the meta-analysis was conducted using a random-effects model. The results showed that bed rest time was shorter in the PTED than the fenestration group (MD = −5.00, 95%CI −6.27 to −3.73, *p* < 0.00001), and the specific results are shown in Appendix A.

#### 3.4.5. Hospitalization Time (Day)

Ten articles [13,27,30,31,32,33,34,36,37,48] reported the hospitalization time in the PTED and MED groups, with 449 and 782 subjects, respectively. The heterogeneity was high (*p* < 0.00001, I^2^ = 95%), and the meta-analysis was conducted using a random-effects model. The results showed that the length of stay was shorter in the PTED group than the MED group (MD = −2.42, −3.21 to −1.63, *p* = 0.0001), and the specific results are shown in Appendix A. Twelve articles [2,6,32,34,38,39,40,42,44,45,46,47] reported the hospitalization time in the PTED and traditional fenestration groups, with 536 and 617 subjects, respectively. The heterogeneity was high (*p* < 0.00001, I^2^ = 96%), and the meta-analysis was conducted using a random-effects model. The results showed that length of stay was shorter in the PTED than the fenestration group (MD = −6.38, −7.69 to −5.08, *p* < 0.00001), and the specific results are shown in Appendix A.

#### 3.4.6. Leg Pain

Eleven articles [6,13,27,29,30,31,32,48,49,50,51] reported the VAS score for leg pain in the PTED and MED groups, with 606 and 881 subjects, respectively. The heterogeneity was high (*p* = 0.03, I^2^ = 51%), and the meta-analysis was conducted using a random-effects model. The results showed that the VAS scores for PTED and MED after the operation were similar (MD = −0.23, −0.61 to 0.15, *p* = 0.60), as shown in Appendix A. Six articles [2,6,32,40,41,52] reported VAS scores for legs after the operation in the PTED and traditional fenestration groups, with 568 and 639 subjects, respectively. The heterogeneity was low (*p* = 0.22, I^2^ = 29%), and the meta-analysis was performed using a fixed-effects model. The results showed that the VAS score in the PTED was similar to that in the traditional fenestration group (MD = −0.05, −0.13 to 0.04, *p* = 0.28), as shown in Appendix A.

#### 3.4.7. Low Back Pain

Nine articles [6,13,27,29,30,31,32,49,51] reported the VAS score for the low back pain in the PTED and MED groups, with 554 and 818 subjects, respectively. The heterogeneity was high (*p* < 0.00001, I^2^ = 90%), and the meta-analysis was performed using a random-effects model. The results showed a lower VAS score for the back pain in the PTED than the MED group (MD = −0.49, −0.84 to −0.14, *p* = 0.006), as shown in Appendix A. Six articles [2,6,32,40,46,52] reported the VAS score for lower back pain in the PTED and traditional fenestration group, with 568 and 639 subjects, respectively. The heterogeneity was high (*p* < 0.00001, I^2^ = 98%), and the meta-analysis was conducted using a random-effects model. The results showed that the VAS score in the PTED was similar to that in the traditional fenestration group after the operation (MD = 0.30, −0.28 to 0.88, *p* = 0.31), as shown in Appendix A.

#### 3.4.8. Postoperative Comprehensive Pain

Five articles [30,42,43,44,47] reported the postoperative comprehensive VAS score in the PTED and traditional fenestration groups, with 164 and 153 subjects, respectively. The heterogeneity was high (*p* < 0.00001, I^2^ = 93%), and the meta-analysis was conducted using a random-effects model. The results showed a similar VAS score for comprehensive pain in the two groups (MD = −0.81, −1.71 to 0.08, *p* = 0.07), as shown in Appendix A.

#### 3.4.9. Postoperative ODI Index

Thirteen articles [13,27,28,29,30,31,32,35,36,37,48,49,51] reported the postoperative ODI index in the PTED and MED groups, with 695 and 992 subjects, respectively. The heterogeneity was high (*p* < 0.00001, I^2^ = 88%), and the meta-analysis was conducted using a random-effects model. The results showed that the ODI index in the PTED group was lower to that in the MED group (MD = −2.21, −4.17 to −0.25, *p* = 0.03), as shown in Appendix A. Nine articles [2,6,32,38,40,44,45,46,52] reported the postoperative ODI index in the PTED and traditional fenestration groups, with 659 and 728 subjects, respectively. The heterogeneity was high (*p* = 0.002, I^2^ = 68%), and the meta-analysis was conducted using a random-effects model. The results showed a similar ODI index in the two groups (MD = −0.62, −1.25 to 0.01, *p* = 0.05), as shown in Appendix A.

#### 3.4.10. Incidence of Complications

Thirteen articles [25,26,28,29,30,32,33,34,37,48,49,50,51] reported the incidence of complications in the PTED and MED groups, with 920 and 1452 subjects, respectively. The heterogeneity was low (*p* = 0.65, I^2^ = 0%), and the meta-analysis was performed using a fixed-effects model. The results showed a similar complication rate between PTED and MED (OR = 0.94, 0.67 to 1.32, *p* = 0.71), as shown in Appendix A. Ten articles [2,28,34,39,40,42,44,47,52,53] reported the incidence of complications in the PTED and traditional fenestration groups, with 518 and 462 subjects, respectively. The heterogeneity was low (*p* = 0.93, I^2^ =0%), and the fixed-effects model for the meta-analysis showed that the incidence of PTED was lower than that of the traditional fenestration group (OR = 0.81, 0.50 to 1.31, *p* = 0.02), as shown in Appendix A.

#### 3.4.11. Incidence of Recurrence

Fifteen articles ([13,25,26,27,28,29,30,31,32,33,34,37,48,50,51] reported the incidence of residual and recurrence in the PTED and MED groups, with 987 and 1634 subjects, respectively. The heterogeneity was low (*p* = 0.93, I^2^ = 0%), and the meta-analysis was performed using a fixed-effects model. The results showed a higher recurrence rate in the PTED than the MED group (OR = 1.55, 1.07 to 2.24, *p* = 0.02), as shown in Appendix A. Seven articles [2,32,34,40,46,52,53] reported the recurrence and residual rate in the PTED and traditional fenestration groups, with 591 and 642 subjects, respectively. The heterogeneity was low (*p* = 0.11, I^2^ = 41%), and the meta-analysis was performed using a fixed-effects model. The results showed a higher recurrence rate in the PTED than the traditional fenestration group (OR = 1.72, 1.02 to 2.90, *p* = 0.04), as shown in Appendix A.

#### 3.4.12. Revision Rate

Eleven articles [13,25,26,28,30,31,32,48,49,50,51] reported the revision rate in the PTED and MED groups, with 868 and 1358 subjects, respectively. The heterogeneity was low (*p* =0.89, I^2^ = 0%), and the meta-analysis was performed using a fixed-effects model. The results showed that PTED had a higher surgical revision rate than MED (OR = 1.67, 1.17 to 2.36, *p* = 0.004), as shown in Appendix A. Four articles [32,46,53,54] reported the revision rate in the PTED and OD groups, with 322 and 589 subjects, respectively. The heterogeneity was low (*p* = 0.98, I^2^ = 0%), and the meta-analysis was performed using a fixed-effects model. The results showed that PTED had a higher surgical revision rate than OD (OR = 2.15, 1.25 to 3.70, *p* = 0.16), as shown in Appendix A.

#### 3.4.13. Rates of Successful Operation

Three articles [26,35,36] reported the successful rate of the operation for PTED and discectomy, with 361 and 589 subjects, respectively. The heterogeneity was low (*p* = 0.98, I^2^ = 0%), and the meta-analysis was performed using a fixed-effects model. The results showed that the PTED was similar to the MED group in terms of successful rate of the operation (OR = 0.64, 0.33 to 1.25, *p* = 0.16), as shown in Appendix A. Five articles [38,41,44,47,52] reported the successful rate of the operation in the PTED group and fenestration group, with 321 subjects in the foraminal PTED and 303 subjects in the fenestration group. The heterogeneity was low (*p* = 0.76, I^2^ = 0%), and the meta-analysis was performed using a fixed-effect model. The results showed that PTED and fenestration had a similar excellence rate of the operation (OR = 0.75, 0.43 to 1.33, *p* = 0.33), as shown in Appendix A.

### 3.5. Sensitivity Analysis

To confirm the stability of the conclusions, a sensitivity analysis was carried out on the thirty-three included articles. There were three articles [26,32,54] with a large sample size (*n* > 200). Two articles [45,47] had a relatively small sample size (*n* < 20). Two RCT studies had a low quality among the literature [44,47]. We excluded these studies, and the remaining literature was again subjected to meta-analysis. In terms of the average operation time, hospital stay, incision length, intraoperative blood loss and VAS pain score (leg and waist), the incidence of complications and the excellent rate of operation, after removing these results, there was no difference in the combined results of meta-analysis. In the sensitivity analysis, the majority of the results were similar to previous findings after excluding the literature with larger or smaller samples with lower quality. This result indicates that the stability of the meta-analysis was relatively high. A few evaluation indexes, such as the ODI of the PTED and MED groups (OR = −2.01, −4.08 to 0.06, *p* = 0.06, I^2^ = 87%) and revision rate of the PTED and traditional open surgery groups (OR = 1.62, 0.54 to 4.82, *p* = 0.39 I^2^ = 0%), were slightly different from the former conclusion, potentially because the number of studies included in the relevant evaluation indexes was relatively small, and the removal of a relatively large sample size may have affected the results. Overall, the results of the meta-analysis were relatively stable.

### 3.6. Publication Bias Analysis

To assess the extent of publication bias, we used funnel plots. Due to the small number of studies in some research indicators, the use of funnel plots to evaluate publication bias can lead to errors. Therefore, the results of the evaluation indexes for groups of no less than five are indicated by funnel plots. We examined the corresponding funnel plots of the various research indicators, which were roughly symmetrical in the visual sense, suggesting that the probability of publication bias was relatively small. The specific publication bias funnel plots are shown in Appendix A.

## 4. Discussion

The incidence of lumbar disc herniation is increasing. For more serious lumbar disc herniation, conservative treatment usually has difficulty achieving satisfactory results. Therefore, surgery is the only way to cure such disorders. Traditional open surgery of the lumbar spine is very traumatic for patients. Thus, clinicians have been trying to find a more minimally invasive procedure that allows a quicker recovery from surgery. The development of the endoscope in the field of surgery has allowed orthopaedic surgeons to be hopeful.

The introduction of PTED has caused great concern [55]. The patients obviously feel the relief of symptoms after the operation. Simultaneously, its minimally invasive advantages greatly shorten the time of hospitalization and reduce patient pain to allow a more rapid return to life and work. Many advantages have led to the rapid development of this technique in China. However, as the number of operations has increased, surgeons have gradually realized that no surgical approach is perfect. PTED is no exception. Studies have shown that this technique is similar to traditional surgery in terms of the efficacy and safety of the surgery [11], but the recurrence rate of PTED is significantly higher than that of open surgery. Lumbar discectomy has been assessed in traditional open surgery, MED and PTED, with decrements of trauma, but whether the effects are increasingly superior, especially the long-term curative effects, has received little agreement.

To determine whether PTED can completely replace MED and traditional open surgery, and thus become the standard procedure for the treatment of lumbar intervertebral disc herniation, we compared the advantages and disadvantages of these three operation methods for the treatment of lumbar intervertebral disc herniation. The main findings of this study are summarized in the following sections.

### 4.1. PTED vs. MED

In comparisons of PTED with MED in terms of surgical indicators, in PTED the incisions were smaller, there was less loss of blood, the postoperative bed time was shorter, and the hospital time was shorter, demonstrating certain advantages of PTED compared with MED. In terms of postoperative efficacy, the ODI index, VAS score for pain in the leg, complication rate of the operation and optimal rate of the operation, there were no statistically significant differences between the two groups. For the VAS score for back pain, the PTED group performed slightly better than the MED group. The advantage of MED was that the postoperative recurrence rate of LDH and revision rate was lower than that of PTED. The specific analysis was as follows.

In terms of surgical indicators, PTED showed advantages mainly in the following aspects. (1) The incision was small, usually approximately 8 mm, and expansion of the channel occurred through the blunt separation of muscle tissue, with significantly less damage to the surrounding soft tissue and muscle than in MED, and thus less blood loss. (2) PTED was performed under anaesthesia [56]. During the operation, the patient remained awake, which made the operation of nerve root decompression safer, and the patients could feel obvious pain relief in the waist and legs. In the early postoperative period, the patients were able to move to the ground, greatly reducing the recovery time, and therefore the length of hospital stay was shorter than that in the MED group. (3) In terms of operation time, although no significant differences were observed between the two groups, in parts of the literature [28,30,34,35,36], the operation time was slightly longer in the PTED than the MED group. The surgical incision of PTED may be small and the operation channel complicated. Concomitantly, when operating under an endoscope, the anatomical structure is quite different compared with its appearance when viewed with only the eyes. All these factors resulted in a slightly longer operation time than in the MED group.

In terms of the postoperative efficacy, the ODI index and VAS score for leg pain were not statistically significant. However, most of the research groups [13,27,29,30,36,49] showed a slightly lower ODI index in the PTED than the MED group, while the VAS score of patients with lower back pain was slightly better in the PTED group than the MED group. This result may be because the ODI index reflects the stability of the lumbar spine after surgery [57]. The VAS score [58] was used to evaluate the improvement of lumbar leg pain in patients with LDH, and it was also an indirect reflection of the degree of surgical trauma. PTED utilized the natural passage into the intervertebral space to remove the protruding nucleus pulposus. The structure of the lumbar spine and the muscles and ligaments in the back were not damaged during the operation [59]. However, MED requires the removal of a small portion of the lamina and yellow ligament, which may have a certain effect on the stability of the lumbar spine. Therefore, the ODI index in the PTED group was slightly lower. Concurrently, the overall damage was small in the PTED group, which could also explain its advantages in the postoperative VAS back pain score. In addition, there were lower incidences of recurrence and revision rates in the MED groups. This result may be because recurrence and revision are caused by the presence of residual disc tissue. Compared with PTED, MED resulted in more complete removal of intervertebral discs. Because PTED can only remove obvious protruding intervertebral disc tissue, it is very difficult for PTED to completely eliminate the remnants of the nucleus pulposus. In addition, the healing scar tissue of the fibrous ring rupture is relatively weak after surgery. Residual disc tissue can easily break through this weak spot, which is another reason for the high recurrence rate.

### 4.2. PTED vs. OD

Compared with the traditional open surgery group, the results for the PTED group showed, in terms of operation indicators, a small incision, less intraoperative blood loss, and a shorter postoperative bed time and length of hospital stay. There were no significant differences in surgical time. For postoperative efficacy, postoperative ODI index, postoperative VAS leg pain, lower back pain score, surgical complication rate, and surgical superior rate, there were no statistically significant differences between the two groups. In terms of recurrence rates and revision rates, open surgery was significantly better than PTED. The specific analysis was as follows.

For operation indicators, less traumatic PTED had obvious advantages, potentially because (1) in traditional open surgery, whether total laminectomy, semi-laminectomy, TLIF, PLIF, or MIS-TLIF, incisions are greater than the 0.8 cm in PTED surgery; (2) in addition, open surgery inevitably destroys the vertebral body, and the muscles and other soft tissues adjacent to the vertebral body need to be detached. However, PTED was inserted through the natural channel to extract the protruding disc, and the bipolar radiofrequency was used during the operation to stop the bleeding. Therefore, the intraoperative blood loss must be much less than in open surgery; (3) PTED was performed under anaesthesia, while traditional open surgery must be performed under general anaesthesia with more loss of blood and more damage to surrounding tissues. Therefore, the postoperative recovery of the patients was slow, and the postoperative bed time was significantly longer than that in the PTED group. The overall length of hospital stay was also extended. Concurrently, a series of complications caused by a long-term bedridden condition, such as pneumonia, lower limb vein thrombosis, and urinary tract infection, also increased the likelihood of occurrence; (4) there were no statistically significant differences in the time of operation, consistent with the research results of Song HP et al. [35]. However, most of the study groups showed that the open group had a slightly longer operative time than the PTED group, potentially due to the complex separation and the need to close the incision layer by layer after completion of the operation, which is likely to increase the operation time.

In terms of surgical efficacy, the VAS score for waist and leg pain, ODI index, and excellence rate of the operation were not statistically significant, similar to the results obtained by Choi et al. [60]. The reason for this result may be because the two operation methods are different. In PTED surgery, the C-arm X-ray is used for positioning, and then the work channel is placed step by step through the intervertebral foramen into the protruding disc [61]. Traditional open surgery is performed from behind. We need to slice through the skin and subcutaneous tissue layer by layer and peel off the tissue next to the vertebral body to expose the protruding intervertebral space. Concurrently, we must also remove some of the lamina and yellow ligament. This process is performed under direct vision. The two operate in different ways, but both are based on the removal of prominent intervertebral discs to relieve the mechanical compression of the nerve root and then to relieve the patient’s waist and leg pain [62]. Therefore, there was no statistically significant difference in postoperative efficacy indicators. The common postoperative complications of surgery are postoperative infection, epidural injury, and nerve root injury [26]. In this study, there were no statistically significant differences in the incidence of complications, potentially because in traditional open surgery, the surgical area must be clearly shown. The operation, in direct view, greatly reduced the damage to the surrounding tissue structures, such as the nerve roots and dural sac. Although the PTED is not performed under direct view, the patient is under anaesthesia. Therefore, if the patient’s nerve root is mildly stimulated during surgery, there will be obvious pain feedback, reminding the surgeon to avoid damage to the nerve roots, among other things. The incidence of infection requires larger samples to explore the causes. In terms of postoperative recurrence and revision rates, open surgery had certain advantages, and the PTED group was relatively higher. The main reason for this finding is that PTED is performed in a non-direct manner. The operation instrument and blind area of the vision make it difficult to fully expose the lamina, nerve root and lateral recess, leading to an incomplete protrusion of the intervertebral disc tissue [59]. Traditional open surgery is performed under direct vision. The anatomical structure of each layer is clearly exposed during the operation, so it is possible to completely eliminate the intervertebral disc tissue. Additionally, the learning curve for PTED is very steep [63]. Although not much time is needed to expose the tissue and suture, the anatomical structure is different under direct vision, and the operating space is very small. For a variety of reasons, doctors who have performed open surgery for many years can take their time. Therefore, lack of experience is also an important factor leading to incomplete removal of the intervertebral disc tissue.

### 4.3. Discussion Regarding the Application of PTED in China

This study included 33 articles, including 19 studies in China and only 14 studies in Europe, the United States, Japan and South Korea. China has a large number of articles about PTED, which indicates that it is well-developed and applied in China, but its operation is not very standard. In contrast, it is used less often abroad. MED and traditional open surgery are the classic surgical methods for the treatment of lumbar disc herniation. Their development occurred much earlier than PTED, and both have satisfactory therapeutic effects after the operation. While the development of PTED was relatively late [40], since its introduction in Beijing 306 Hospital in the 1990s, it has become the mainstream technology to treat lumbar disc herniation. It originated in Europe and America and then developed quickly in east Asia, especially in China.

Why is it so popular in China and less applied in related fields abroad?

The reasons are as follows. The medical community in China has a common characteristic, which is to promote new technologies. Simultaneously, they are good at achieving greater advantages. Currently, lumbar disc herniation is one of the most common diseases in spine surgery, its incidence is increasing annually, and the age of onset is getting younger [64]. Additionally, most Chinese people have more or less social or family responsibilities. The patient’s mind is concerned with early postoperative activities to resume daily work as soon as possible. China has a large patient population, and their desire to recover early is very strong. As a new minimally invasive operation, the PTED incision is only 8 mm. The trauma is small during the operation, and the postoperative recovery is quick. Chinese doctors can use this as a gimmick to attract more lumbar disc herniation patients to perform PTED.

2.Defects of PTED

Although PTED is widely used in China’s clinical field, its disadvantages are also obvious: 1. The cost of PTED is far greater than MED and traditional open surgery. Zihao Chen et al. [30]. reported that PTED had a significantly higher cost than MED. If the cost of revision is considered, the total cost becomes enormous. Although China has developed rapidly in recent years, most patients are still relatively poor. The reimbursement rate for China’s medical insurance is not very high. The cost of the operation is so great that it is difficult for the patient to accept. Additionally, China’s health reform has become imperative. The high cost of PTED is not conducive to the cost control of Chinese medical reform. 2. The radiation exposure associated with PTED greatly exceeds that of MED and traditional open surgery. Peng Zhou et al. [42]. reported that the amount of radiation during the operation was significantly higher than traditional open surgery. The reason for this difference is simple. PTED is very demanding in terms of targeting localization. Doctors need an accurate position and puncture before surgery, which relies heavily on X-ray fluoroscopy [65]. Ahn et al. [66]. first quantified the radiation dose of a surgeon in 30 cases of the operation. They concluded that orthopaedic surgeons who do not take protective measures can easily exceed the radiation limits of professionals. Radiation exposure has a significant impact on both the patient and the surgeon’s health. Therefore, the high radiation exposure of PTED should be paid more attention by Chinese physicians. 3. The health economy of PTED is not high. From the perspective of health economics, we apply PTED to the clinic, but its input and output economic proportions are low, which is unfavourable for its long-term development in the medical field. 4. The high cost of PTED. As an emerging high-technology equipment, the research and development cost of PTED is relatively high, so its cost is correspondingly high. Without investment, it is relatively difficult for local hospitals to introduce this technology [67]. Consequently, patients will be gathered at large hospitals for treatment. As lumbar disc herniation is a common, uncomplicated clinical disease, it is bound to take up a large amount of high-quality medical resources, in contrast to the “graded diagnosis and treatment” policy vigorously promoted by the medical community in China today [68]. 5. The learning curve of PTED is very steep. There are currently many young doctors in China’s medical system. The small operation space of PTED and the difference in anatomical structures with direct vision make it more difficult to master this technique compared with the other two operations. It is difficult for doctors without years of open surgery experience to grasp the operation process of PTED, which is extremely unfavourable for young doctors in the current medical environment in China. 6. High recurrence rate and high revision rate of PTED. The definition of the postoperative recurrence of PTED is not completely unified, and Hoogland et al. reported that the recurrence rate of PTED for the treatment of LDH within 1 year was approximately 3.92% [69]. Because PTED cannot completely clear the prominent nucleus pulposus, the postoperative recurrence and residual rate are greater than those of MED and open surgery. The incidence of long-term complications is also relatively high. Patients were not satisfied with the postoperative outcome, and many of them even needed renovations. Under the background of the Chinese medical industry, an operation not being effective or requiring renovation are both taboo. To a certain extent, the unsuccessful operation is bad for the doctor’s career. Considering the defects of PTED, Chinese doctors must consider its application seriously. It is not appropriate to focus solely on immediate interests, and we should not blindly consider replacing MED and traditional open surgery with PTED.

### 4.4. Strengths and Weaknesses

The advantages of this study include the following aspects. (1) This meta-analysis synchronously compared PTED with MED and PTED with traditional open surgery. Thus, it can be stated that all the major surgical procedures for the treatment of lumbar disc herniation were examined. (2) In this study, various indexes of different surgical procedures were compared in all directions, including relevant indicators in the surgical procedure and the postoperative curative effect. Simultaneously, we analysed the reasons for the differences objectively, leading to a high reference. (3) For many studies of PTED in China, we have consulted a large number of related literatures and mainly discussed the reasons for its rapid development in China. Concurrently, we analysed the disadvantages of the blind application of PTED from multiple angles. (4) In this study, compared with other former related meta-analyses, we conducted a sensitivity analysis of the combined results to further investigate the stability of the results. The funnel plot was used to analyse publication bias, increasing the credibility of the research results.

This study also has the following shortcomings. (1) Due to the limited number of controlled trials comparing PTED with other surgical methods in the database, relatively few studies were included in this meta-analysis. In addition, except for several large-sample studies, most of the sample sizes were relatively small. (2) Most of the average follow-up times included in this study were 1 to 2 years. The long-term effects of the surgical approach will require a longer follow-up. (3) Randomized controlled clinical trials are relatively difficult to implement in the field of surgery, so most of the articles in this study were retrospective studies. In addition, some of the RCT studies included in the study were unknown in terms of random sequence generation, and whether the blinded method was adopted was not reported, which may have resulted in a certain bias in the study results [70]. (4) Some of the evaluation indexes in the study had high heterogeneity [71], potentially due to a defect in the research. Because of the clinical heterogeneity, different regions, different operation techniques, different proficiencies of the operators, and different condition extents of the patient, a variety of factors can lead to great heterogeneity in operation indicators. Further subgroup or regression analysis to explore the heterogeneity source is more complex. Therefore, subgroup analysis was conducted only for regional differences. (5) The use of a funnel plot to evaluate the publication bias of the study results is relatively subjective, so the results of the publication bias analysis are presented only for reference [72,73]. (6) The pre-operative herniation characteristic, neurological deficit, and duration of symptoms were not specified in most of the included studies, but the type of disc herniation and the severity and duration of pain symptoms have a crucial impact on the difficulty of surgery and the efficacy after surgery, so it is also one of the shortcomings of this study. In the future, we can explore the influence of the severity of disc herniation and the duration of symptoms on different surgical methods. (7) At present, endoscopic spine surgery is mainly divided into two approaches, the earliest transforaminal approach (PTED), and the later-developed translaminar approach (PEID). [74]. Different surgical approaches have different effects on surgery-related indicators and postoperative efficacy. This study only discusses the most common PTED surgery, and the indications for PEID are relatively narrow. In the future, the relevant indicators of PEID and open surgery can be compared and studied.

## 5. Conclusions

In summary, compared with MED and traditional open surgery, the microinvasive advantages of PTED are obvious. However, in terms of its postoperative efficacy, there were no significant differences between PTED and the other two treatment options. In addition, the postoperative recurrence rate and “revision” rate were slightly higher for PTED than that of traditional open surgery and MED. This finding is closely related to the incomplete removal of the intervertebral disc during PTED, which underscores a direction for the improvement of PTED technology. The development of PTED in China has been much more rapid than that in its regions of origin. However, it is still difficult to cover its disadvantages: the high incidence of long-term complications, postoperative recurrence, renovation, and exorbitant surgical costs. Additionally, its economic health is lower, which indicates poor prospects for health-care reform to control costs. The rate of exposure is also high with PTED, which has negative impacts on doctors and patients. In conclusion, the advantage of PTED lies in its minimally invasive nature, which also shows its flaws. If we want to replace MED or traditional open surgery with PTED, we need to develop and refine this surgical approach.

This study, as discussed earlier, has many limitations. Thus, to gain more reliable evidence, additional trials with higher quality, a longer follow-up time, larger sample sizes, and more randomized controlled trials are needed to evaluate the long-term efficacy of different surgical methods, complications and recurrence rates for the treatment of patients with lumbar disc prolapse to provide more reliable information.

## Figures and Tables

**Figure 1 jcm-11-06604-f001:**
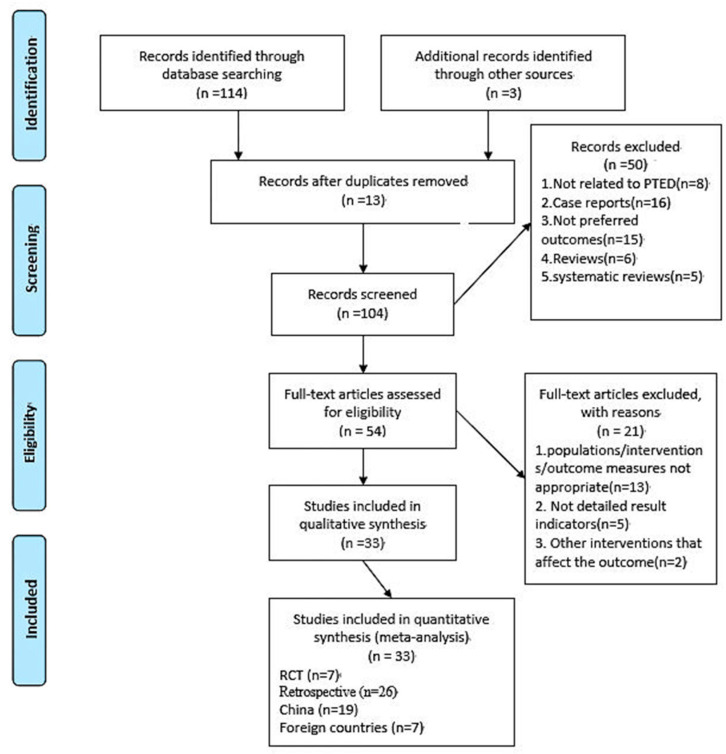
Flow chart of study selection.

**Table 1 jcm-11-06604-t001:** Basic information of the studies.

Article (Author and Year)	Study Type	Operation Method	Number of Patients	Age (Year)	Follow-Up Time (Month)	Outcome Indicators
Ma et al., 2022	Retrospective	PTED	46	45.80 ± 17.15	24	8 9 10 11
		OD	45	40.49 ± 15.21	24	
Wang et al., 2021	Retrospective	PTED	69	51.25 ± 17.26	6	1 2 5 6 7 9 10
		MED	86	47.26 ± 16.49	6	
Rajamani et al., 2021	Retrospective	PTED	86	39.4 ± 12.38	24	1 2 3 5 6 7 10
		MED	351	44.3 + 12.76	24	
		OD	145	37.6 + 7.39	24	
Jing et al., 2021	Retrospective	PTED	31	51.32 + 8.99	27.71 + 1.92	1 2 3 5 8 9
		MED	31	50.19 + 9.36	27.26 + 1.53	
Jarebi et al., 2021	Retrospective	PTED	29	47.21 ± 12.55	24	5 6 7 8 9 10
		MED	29	46.97 ± 12.55	24	
Meyer et al., 2020	RCT	PTED	23	47.2 ± 10.6	12	6 8 9 10
		MED	24	45.2 ± 10.6	12	
Chen et al., 2020	RCT	PTED	119	40.9 + 11.9	24	6 7 8 9 10
		MED	122	41.0 + 10.8	24	
Kim et al., 2019	Retrospective	PTED	173	46.49 + 14.28	120	10
		OD	1683	46.49 + 14.28	120	
Ahn et al., 2019	Retrospective	PTED	146	32.7 (16–70)	60	1 5 6 7 9 10
		OD	152	35.4 (14–77)	60	
Chen et al., 2018	RCT	PTED	80	40 ± 11.4	12	1 4 5 6 7 8 9 10
		MED	73	40.7 ± 11.1	12	
Liu et al., 2018	Retrospective	PTED	60	36.2 ± 5.9	28.2 ± 2.5	1 2 3 6 7 8 9
		MED	63	33.1 ± 6.7	29.6 ± 3.7	
Gibson et al., 2016	RCT	PTED	70	42 ± 9	24	6 7 8 10
		MED	70	39 ± 9	24	
Sun et al., 2017	Retrospective	PTED	11	54.91 ± 8. 28	12	1 2 4 5 6 7
		OD	13	58.08 ± 11.30	12	
Ahn et al., 2015	Retrospective	PTED	32	22.41 ± 1.68	13.69 ± 1.26	1 5 6 7 8 9
		OD	34	22.18 ± 1.51	13.41 ± 1.02	
Pan et al., 2016	Retrospective	PTED	48	39.5	12	1 2 5 6 7
		OD	58	42.8	12	
Ariun et al., 2015	Retrospective	PTED	36	44.17 ± 6.54	12	1 5 7 11
		MED	50	41.46 ± 7.22	12	
Hsien-TA et al., 2012	Retrospective	PTED	57	44.2	20.4 (12–24)	1 7 8 9 10
		MED	66	50.4	20.4 (12–24)	
Lee et al., 2009	Retrospective	PTED	25	42.0 ± 11.4	34.0 ± 4.4	1 5 6 7 8 9
		MED	29	47.7 ± 12.2	34.3 ± 4.6	
Li et al., 2018	Retrospective	PTED	48	18.96 ± 1.99	68.87 ± 7.03	1 2 5 6 7 9
		MED	30	19.40 ± 1.50	67.07 ± 6.76	
Ding et al., 2017	RCT	PTED	50	41.32 ± 11.53	41.32 ± 11.53	1 3 4 5 6 7 11
		OD	50	43.90 ± 11.8	43.90 ± 11.8	
Chang et al., 2017	Retrospective	PTED	60	52.54 ± 4.12	52.54 ± 4.12	1 2 3 4 6
		OD	50	53.67 ± 4.28	53.67 ± 4.28	
Li et al., 2018	Retrospective	PTED	33	43.9 ± 11.6	43.9 ± 11.6	1 2 5 6
		OD	30	46.1 ± 13.2	46.1 ± 13.2	
Jeoug et al., 2006	Retrospective	PTED	22	56 ± 9.12	12	1 6 11
		OD	25	56.45 ± 10.89	12	
Pan et al., 2014	RCT	PTED	10	No discussion	6	2 3 5 6 11
		OD	10	No discussion	6	
Song et al., 2017	Retrospective	PTED	30	54.8 ± 6.5	18	1 2 3 4 7 11
		MED	30	53.6 ± 6.4	18	
Liu et al., 2016	Retrospective	PTED	209	57.2	46.5 (12–69)	6 7 9 11
		OD	192	55.9	46.5 (12–69)	
Choi et al., 2016	Retrospective	PTED	20	33.9 ± 11.1	27.5 ± 5.7	1 5 6 7 9 10
		MED	23	38 ± 11.6	27.5 ± 5.7	
Yao et al., 2016	Retrospective	PTED	47	47.91 ± 14.77	47.91 ± 14.77	1 5 6 7 9
		OD	58	46.76 ± 12.37	46.76 ± 12.37	
Kim et al., 2007	Retrospective	PTED	301	34.9	23.6 (18–36)	1 8 9 10 11
		MED	614	44.4	23.6 (18–36)	
Wang et al., 2013	Retrospective	PTED	25	17.9 ± 1.9	No discussion	1 2 5 8 9
		MED	80	17.9 ± 1.9	No discussion	
		OD	16	17.9 ± 1.9	No discussion	
Chen et al., 2015	Retrospective	PTED	18	57.4 ± 12.4	24	1 5 8
		OD	25	54.9 ± 16. 6	24	
Li et al., 2015	Retrospective	PTED	30	17.9 ± 10.4	12	1 2 5 7 11
		OD	26	18.8 ± 10.4	12	
Mayer et al., 1993	RCT	PTED	20	39.8 ± 10.4	24	1 8 9 10
		MED	20	42.7 ± 10	24	

RCT, randomized controlled clinical trial. 1. Average operative time (min). 2. Intraoperative blood loss (mL). 3. Size of incision (cm). 4. Postoperative bed rest time (day). 5. Hospitalization time. 6. VAS score of leg/low back pain. 7. Postoperative ODI index. 8. Incidence of complications. 9. Incidence of recurrence. 10. Revision rate. 11. Excellent rate of operation.

**Table 2 jcm-11-06604-t002:** Quality evaluation of retrospective clinical studies.

Included Studies (Author and Year)	Study Type	Rated Items								Quality Score
		Selection (4)				Comparability (2)	Outcomes (3)			
		①	②	③	④	⑤	⑥	⑦	⑧	
Ma et al., 2022	Retrospective	1	1	1	1	2	1	1	1	9 (Hi-Q)
Wang et al., 2021	Retrospective	1	1	1	1	2	1	1	1	9 (Hi-Q)
Rajamani et al., 2021	Retrospective	1	1	1	1	2	1	1	1	9 (Hi-Q)
Jing et al., 2021	Retrospective	1	1	1	1	1	1	1	1	8 (Hi-Q)
Jarebi et al., 2021	Retrospective	1	1	1	1	2	1	1	1	9 (Hi-Q)
Kim et al., 2019	Retrospective	1	1	1	1	1	1	1	1	8 (Hi-Q)
Ahn et al., 2019	Retrospective	1	1	1	1	2	1	1	1	9 (Hi-Q)
Liu et al., 2018	Retrospective	1	1	1	1	2	1	1	1	9 (Hi-Q)
Sun et al., 2017	Retrospective	1	1	1	1	2	1	1	1	9 (Hi-Q)
Ahn et al., 2015	Retrospective	1	1	1	1	2	1	1	1	9 (Hi-Q)
Pan et al., 2016	Retrospective	1	1	1	1	2	1	1	1	9 (Hi-Q)
Ariun et al., 2015	Retrospective	1	1	1	1	2	1	1	1	9 (Hi-Q)
Hsien-Ta et al., 2012	Retrospective	1	0	0	1	1	1	1	1	6 (Hi-Q)
Lee et al., 2009	Retrospective	1	1	1	1	2	1	1	1	9 (Hi-Q)
Li et al., 2018	Retrospective	1	1	1	1	1	1	1	1	8 (Hi-Q)
Chang et al., 2017	Retrospective	1	1	1	1	1	1	1	1	8 (Hi-Q)
Li et al., 2018	Retrospective	1	1	1	1	2	1	1	1	9 (Hi-Q)
Jeoug et al., 2006	Retrospective	1	1	1	1	1	1	1	1	8 (Hi-Q)
Song et al., 2017	Retrospective	1	1	1	1	2	1	1	1	9 (Hi-Q)
Liu et al., 2016	Retrospective	1	1	0	1	2	1	1	1	9 (Hi-Q)
Choi et al., 2016	Retrospective	1	1	1	1	1	1	1	1	8 (Hi-Q)
Yao et al., 2016	Retrospective	1	1	1	1	2	1	1	1	9 (Hi-Q)
Kim et al., 2007	Retrospective	1	1	1	1	1	1	1	1	8 (Hi-Q)
Wang et al., 2013	Retrospective	1	1	0	1	1	1	1	1	7 (Hi-Q)
Chen et al., 2015	Retrospective	1	1	1	1	2	1	1	1	9 (Hi-Q)
Li et al., 2015	Retrospective	1	1	0	1	1	1	1	1	7 (Hi-Q)

①: Case definition; ②: representativeness: ③; control selection; ④: control definition; ⑤ A/B: the comparability of cases and controls in in design and statistical analysis; ⑥: ascertainment of exposure; ⑦: same method of ascertainment for cases and controls; ⑧: non-response rate.

**Table 3 jcm-11-06604-t003:** Quality evaluation of randomized controlled clinical trials.

Studies (Author and Year)	Study Type	Randomization (2)	Concealment of Allocation (2)	Double Blinding (2)	Withdrawals and Dropouts (1)	Quality Evaluation
Meyer et al., 2020	RCT	2	1	0	1	4 (high quality)
Chen et al., 2020	RCT	2	2	2	1	7 (high quality)
Chen et al., 2016	RCT	2	2	2	1	7 (high quality)
Gibson et al., 2016	RCT	2	2	1	1	7 (high quality)
Ding et al., 2017	RCT	1	0	0	1	3 (low quality)
Pan et al., 2014	RCT	1	1	0	1	3 (low quality)
Mayer et al., 1993	RCT	2	1	0	1	4 (Igh quality)

**Table 4 jcm-11-06604-t004:** Meta-analysis for surgical related indicators and postoperative curative effect.

Outcome	Control Group	Treatment Effects Difference	*p* for Treatment Effects Difference	I^2^ for Heterogeneity	*p* for Heterogeneity
Average operative time (min)	PTED vs. MED	MD = −8.74, −24.15 to 6.66	*p* = 0.27	I^2^ = 99%	*p* < 0.00001
	PTED vs. OD	MD = −22.50, −41.03 to −3.98	*p* = 0.02	I^2^ = 99%	*p* < 0.00001
Intraoperative blood loss (ml)	PTED vs. MED	MD = −24.96, −34.12 to −15.81	*p* < 0.00001	I^2^ = 99%	*p* < 0.00001
	PTED vs. OD	MD = −89.29, −111.80 to −66.79	*p* <0.00001	I^2^ = 99%	*p* < 0.00001
Size of the incision (cm)	PTED vs. MED	MD =−1.56, −2.63 to −0.50	*p* = 0.004	I^2^ = 100%	*p* < 0.00001
	PTED vs. OD	MD = −3.11, −3.99 to −2. 22	*p* < 0.00001	I^2^ = 99%	*p* < 0.00001
Postoperative bed rest time (day)	PTED vs. MED	MD = −2.45, −4.13 to −0.76	*p* = 0.004	I^2^ = 97%	*p* < 0.00001
	PTED vs. OD	MD = −5.00, −6.27 to −3.73	*p* < 0.00001	I^2^ = 91%	*p* < 0.0001
Hospitalization time (day)	PTED vs. MED	MD = −2.42, −3.21 to −1.63	*p* < 0.00001	I^2^ = 95%	*p* < 0.00001
	PTED vs. OD	MD = −6.38, −7.69 to −5.08	*p* < 0.00001	I^2^ = 96%	*p* < 0.00001
Leg pain	PTED vs. MED	MD = −0.23, −0.61 to 0.15	*p* = 0.60	I^2^ = 51%	*p* = 0.03
	PTED vs. OD	MD = −0.05, −0.13 to 0.04	*p* = 0.28	I^2^ = 29%	*p* = 0.22
Low back pain	PTED vs. MED	MD = −0.49, −0.84 to −0.14	*p* = 0.006	I^2^ = 90%	*p* < 0.00001
	PTED vs. OD	MD = 0.30, −0.28 to 0.88	*p* = 0.31	I^2^ = 98%	*p* < 0.00001
Postoperative comprehensive pain	PTED vs. OD	MD = −0.81, −1.71 to 0.08	*p* = 0.07	I^2^ = 93%	*p* < 0.00001
Postoperative ODI index	PTED vs. MED	MD = −2.21, −4.17 to −0.25	*p* = 0.03	I^2^ = 88%	*p* < 0.00001
	PTED vs. OD	MD = −0.62, −1.25 to 0.01	*p* = 0.05	I^2^ = 68%	*p* = 0.002
Incidence of complications	PTED vs. MED	OR = 0.94, 0.67 to 1.32	*p* = 0.71	I^2^ = 0%	*p* = 0.65
	PTED vs. OD	OR = 0.81, 0.50 to 1.31	*p* = 0.39	I^2^ = 0%	*p* = 0.47
Incidence of recurrence	PTED vs. MED	OR = 1.55, 1.07 to 2.24	*p* = 0.02	I^2^ = 0%	*p* = 0.93
	PTED vs. OD	OR = 1.72, 1.02 to 2.90	*p* = 0.04	I^2^ = 41%	*p* = 0.11
Revision rate	PTED vs. MED	OR = 1.67, 1.17 to 2.36	*p* = 0.004	I^2^ = 0%	*p* = 0.89
	PTED vs. OD	OR = 2.15, 1.25 to 3.70	*p*= 0.006	I^2^ = 0%	*p* = 0.55
Rates of successful operation	PTED vs. MED	OR = 0.64, 0.33 to 1.25	*p* = 0.16	I^2^ = 0%	*p* = 0.98
	PTED VS OD	OR = 0.75, 0.43 to 1.33	*p* = 0.33	I^2^ = 0%	*p* = 0.76

## Data Availability

The study protocol can be accessed on the International Prospective Register of Systematic Reviews (PROSPERO 2018: CRD42018094890, http://crd.york.ac.uk/PROSPERO accessed on 8 August 2022). Additional data not presented in the manuscript can be obtained by contacting the authors.

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
