# Peer review of "Comparison of Three Common Intervertebral Disc Discectomies in the Treatment of Lumbar Disc Herniation: A Systematic Review and Meta-Analysis Based on Multiple Data"

_jcm, 2022, doi:10.3390/jcm11226604_

Round 1

Reviewer 1 Report

nice manuscript on an actual topic

This manuscript deals with the Comparison of three common intervertebral disc Discectomies in the treatment of lumbar disc herniation. The authors performed a systematic review and meta-analysis of the literature.The surge of the literatur, the selection of the papers and the evaluation of pros and cons are adequate. Hence, the conclusion of the authors is valid. The therapeutic effect and safety of PTED, MED and OD in the treatment of lumbar disc herniation were similar. A reason for that could not be given. PTED had obvious advantages with rapid recovery after surgery, but also short commings like the high recurrence rates and revision rates. No data were given on a possible flat learning curve of the surgeans with PTED. 

Author Response

Responses to the Reviewers

Reviewer#1's comments:

Point1:

This manuscript deals with the Comparison of three common intervertebral disc Discectomies in the treatment of lumbar disc herniation. The authors performed a systematic review and meta-analysis of the literature.The surge of the literatur, the selection of the papers and the evaluation of pros and cons are adequate. Hence, the conclusion of the authors is valid. The therapeutic effect and safety of PTED, MED and OD in the treatment of lumbar disc herniation were similar. A reason for that could not be given.

Response 1:

Thanks for your suggestion. Intervertebral disc herniation(LDH) is the rupture of the annulus fibrosus of the intervertebral disc and the protrusion of the nucleus pulposus tissue from the rupture, which will stimulate or compress the spinal nerve roots and spinal cord, resulting in a series of clinical symptoms such as neck, shoulder, low back and leg pain, numbness and so on. Therefore, whether it is traditional open surgery(OD), relatively minimally invasive microendoscopic discectomy(MED) or minimally invasive transforaminal endoscopic discectomy(PTED), they are all from the perspective of open or minimally invasive, to remove the protruding nucleus pulposus tissue and relieve the related nerve compression, so there is no significant difference in the early efficacy and safety of these surgical methods. Similar conclusions can be drawn from the references included in this study. However, from the perspective of medium and long term efficacy, due to the limitation of operation space, PTED may not completely remove the nucleus pulposus, so there is a risk of high recurrence rate and revision rate.

Point2:

PTED had obvious advantages with rapid recovery after surgery, but also short commings like the high recurrence rates and revision rates. No data were given on a possible flat learning curve of the surgeans with PTED. 

Response 2:

Thank you for your suggestion. Since PTED is a minimally invasive surgery, compared with the direct vision of open surgery(OD), its surgical operation requires better spatial stereoscopic thinking. It is difficult for surgeons without many years of open surgical operation experience to master this technology. Thus, the learning curve for PTED is relatively steep rather than a flat learning curve. This is also the reason for the uneven operation time, postoperative efficacy and other indicators of PTED performed by surgeons at different surgical levels. The research has relevant references to illustrate this point of view.

Reviewer 2 Report

Patients who have undergone percutaneous transforaminal discectomy (PTED) have superiority over MED and OD as a minimally invasive procedure. A systematic review and meta-analysis showed that PTED tends to have higher recurrence rates and revision rates and is less cost-effective than the other two methods. These findings should serve as a warning against easily replacing PTED with MED or OD.

Recently, percutaneous interlaminar discectomy has been reported as an alternative to PTED. Given the similarity of entry approaches, percutaneous interlaminar discectomy should be compared to MED and OD. In the Discussion section, not only PTED but also percutaneous interlaminar discectomy should be introduced. In addition, the reasons for the lack of comparison with interlaminar discectomy should be mentioned as a limitation of this study.

Author Response

Responses to the Reviewers

Reviewer#2's comments:

Point1:

Patients who have undergone percutaneous transforaminal discectomy (PTED) have superiority over MED and OD as a minimally invasive procedure. A systematic review and meta-analysis showed that PTED tends to have higher recurrence rates and revision rates and is less cost-effective than the other two methods. These findings should serve as a warning against easily replacing PTED with MED or OD.

Recently, percutaneous interlaminar discectomy has been reported as an alternative to PTED. Given the similarity of entry approaches, percutaneous interlaminar discectomy should be compared to MED and OD. In the Discussion section, not only PTED but also percutaneous interlaminar discectomy should be introduced. In addition, the reasons for the lack of comparison with interlaminar discectomy should be mentioned as a limitation of this study.

Response 1:

Thank you for your suggestion. Spinal endoscopy is divided into two approaches, the earliest is the transforaminal approach(PTED), and the interlaminar approach(PEID) was developed later. According to the comments of the reviewers, we added the content of surgical methods through interlaminar approach in the introduction and discussion section (see the red section of the revised manuscript for details).

In addition, this study only investigated the transforaminal approach to endoscopic spine surgery, known as PTED, and not the interlaminar approach, known as PEID, because: PEID is not the first choice in the treatment of intervertebral disc herniation, and its indications are relatively narrow, mainly lumbar 4-5, lumbar 5-sacral 1 disc herniation. For patients with higher iliac crest and unable to perform PTED, we consider the application of posterior PEID for treatment. At the same time, PEID is similar to open surgery and microendoscopic discectomy approach, which can be said to be a more minimally invasive small incision microendoscopic discectomy. Therefore, compared with OD and MED, PEID is not as innovative as PTED. However, the reviewer's comments were very pertinent, and we added this part to the limitations of this study (see the red section of the revised manuscript for details).

Reviewer 3 Report

The authors compare 3 techniques for discectomy procedures using metanalysis techniques. 

The article is of importance and well constructed and presented.

The groups of PTED and Open do not seem compared in the current analysis. Were there no adequate studies addressing this?

The limitations of pre-operative herniation characteristic, neurological deficit, and duration of symptoms not being included should be noted. 

At what time point was the post-operative outcomes reported?

The findings of enhanced recovery without control for type and severity of herniation and with current metrics do not seem justified. 

Author Response

Responses to the Reviewers

Reviewer#3's comments:

Point1:

The authors compare 3 techniques for discectomy procedures using metanalysis techniques. 

The article is of importance and well constructed and presented.

The groups of PTED and Open do not seem compared in the current analysis. Were there no adequate studies addressing this?

Response 1:

Thank you for your suggestion. In this study, the operation related indexes and postoperative efficacy of PTED and conventional open surgery(OD) and PTED and endoscopic discectomy(MED) were compared respectively.

Point2:

The limitations of pre-operative herniation characteristic, neurological deficit, and duration of symptoms not being included should be noted. 

Response 2:

Thank you for your suggestion. For lumbar disc herniation(LDH), the type of disc herniation, the preoperative neurological deficit symptoms and their duration have a crucial impact on the difficulty of surgery and the postoperative efficacy. As this part was not detailed in most of the included clinical studies, and our meta-analysis was a secondary data analysis, so the preoperative specific conditions of the patients could not be recorded completely. The reviewer's comments were very good, and we included this part in the discussion section as a limitation of this study (see the red section of the revised manuscript for details).

Point3:

At what time point was the post-operative outcomes reported?

The findings of enhanced recovery without control for type and severity of herniation and with current metrics do not seem justified. 

Response 3:

Thank you for your suggestion. In Table I (Baseline information for enrolled patients), we provide the duration of postoperative follow-up that was explicitly specified in some of the included studies. Postoperative related evaluation indicators are also indicators within this time frame. At the same time, this study only compared the postoperative efficacy indicators, and concluded that there was no significant difference in pain relief and maintenance of lumbar stability compared with open surgery(OD) and microdiscectomy(MED). Similar to the previous question, the type and severity of symptoms of intervertebral disc herniation have a very important impact on the surgical efficacy, but most of the included studies did not elaborate on this, so we also add the comments given by the reviewers in the limitations of this study (see the red section of the revised manuscript for details).

.